# Mesoscale simulations predict the role of synergistic cerebellar plasticity during classical eyeblink conditioning

Alice Geminiani[1¤]*, Claudia Casellato[1,2], Henk-Jan Boele[3,4], Alessandra Pedrocchi[5], Chris I. De Zeeuw[3,6], Egidio D'Angelo[1,2]*

1 Department of Brain and Behavioral Sciences, University of Pavia, Pavia, Italy, 2 Digital Neuroscience Center, IRCCS Mondino Foundation, Pavia, Italy, 3 Department of Neuroscience, Erasmus MC, Rotterdam, The Netherlands, 4 Neuroscience Institute, Princeton University, Washington Road, Princeton, New Jersey, United States of America, 5 NearLab, Department of Electronics, Information and Bioengineering, Politecnico di Milano, Milan, Italy, 6 Netherlands Institute for Neuroscience, Amsterdam, The Netherlands

¤ Current address: Champalimaud Foundation, Lisbon, Portugal
* alice.geminiani@unipv.it (AG); egidiougo.dangelo@unipv.it (EDA)

**Data Availability Statement:** The neuron and synaptic plasticity models implemented as a NEST extension module are available at https://github.

## Abstract

According to the motor learning theory by Albus and Ito, synaptic depression at the parallel fibre to Purkinje cells synapse (*pf*-PC) is the main substrate responsible for learning sensori-motor contingencies under climbing fibre control. However, recent experimental evidence challenges this relatively monopolistic view of cerebellar learning. Bidirectional plasticity appears crucial for learning, in which different microzones can undergo opposite changes of synaptic strength (e.g. downbound microzones–more likely depression, upbound micro-zones—more likely potentiation), and multiple forms of plasticity have been identified, distributed over different cerebellar circuit synapses. Here, we have simulated classical eyeblink conditioning (CEBC) using an advanced spiking cerebellar model embedding downbound and upbound modules that are subject to multiple plasticity rules. Simulations indicate that synaptic plasticity regulates the cascade of precise spiking patterns spreading throughout the cerebellar cortex and cerebellar nuclei. CEBC was supported by plasticity at the *pf*-PC synapses as well as at the synapses of the molecular layer interneurons (MLIs), but only the combined switch-off of both sites of plasticity compromised learning significantly. By differentially engaging climbing fibre information and related forms of synaptic plasticity, both microzones contributed to generate a well-timed conditioned response, but it was the downbound module that played the major role in this process. The outcomes of our simulations closely align with the behavioural and electrophysiological phenotypes of mutant mice suffering from cell-specific mutations that affect processing of their PC and/or MLI synapses. Our data highlight that a synergy of bidirectional plasticity rules distributed across the cerebellum can facilitate finetuning of adaptive associative behaviours at a high spatiotemporal resolution.

com/dbbs-lab/cereb-nest. The simulation and analysis code is available at https://github.com/AliceGem/mesoscale_simulations_cebc. All the other relevant data are within the manuscript and its Supporting Information files.

**Funding:** This research has received funding from the European Union's Horizon 2020 Framework Program for Research and Innovation under the Specific Grant Agreement No. 945539 (Human Brain Project SGA3, Partnering Project "CerebNEST", Voucher No. 49 "Virtual Mouse CerebNEST"), the European Union's Horizon Europe Programme under the Specific Grant Agreement No. 101147319 (EBRAINS 2.0 Project) and from the Italian Ministry of Research through the PNRR projects funded by the European Union – NextGenerationEU "A multiscale integrated approach to the study of the nervous system in health and disease" (Project PE0000012, CUP F13C2200124007, "MNESYS") to ED, "European Brain Research Infrastructure - Italy" (Project IR0000011, CUP B51E22000150006, "EBRAINS-Italy") to AP and "National Centre for HPC, Big Data and Quantum Computing" (Project CN00000013 PNRR MUR – M4C2 – Fund 1.4 - Call "National Centers" - law decree n. 3138 16 december 2021) to CC, and a PhD scholarship (AG). Financial support to CIDZ and HJB was provided by the Netherlands Organization for Scientific Research (NWO-ALW 824.02.001), the Dutch Organization for Medical Sciences (ZonMW 91120067), Medical Neuro-Delta (MD 01092019-31082023), INTENSE LSH-NWO (TTW/00798883), ERC-adv (GA-294775) and ERC-POC (nrs. 737619 and 768914); The NIN Vriendenfonds for Albinism as well as the Dutch NWO Gravitation Program (DBI2). The funders had no role in study design, data collection and analysis, decision to publish, or preparation of the manuscript.

**Competing interests:** The authors have declared that no competing interests exist.

## Author summary

The cerebellum plays a key role in motor learning thanks to synaptic plasticity. While synaptic depression at one cerebellar synaptic type has been long considered the main site for learning, recent experimental findings point out to a critical role for bidirectional plasticity at multiple cerebellar synapses. In this work, we developed a spiking neural network model of the cerebellum and simulated an associative cerebellum-driven task, i.e. classical eyeblink conditioning. Our simulations closely reproduced the behavioural phenotypes of mutant mice with altered cerebellar synapses, shedding light on the underlying neural mechanisms. The results highlight that a synergy of bidirectional plasticity distributed across the cerebellum is necessary for finetuning of adaptive associative behaviours at a high spatiotemporal resolution.

## 1. Introduction

The cerebellum plays a key role in sensorimotor adaptation by learning the precise timing of correlated events and regulating the time and gain of motor responses. This allows the cerebellum to predict the effect of perturbations and coordinate anticipatory movements [1–3]. A benchmark protocol for investigating cerebellar functioning is classical eyeblink conditioning (CEBC). In this simple task, two sensory stimuli are provided in sequence time-locked to each other: a neutral conditioned stimulus (CS, usually a tone or a LED light) followed at a specific Inter-Stimulus Interval (ISI) by an unconditioned stimulus (US, usually an air puff or a periorbital electrical stimulation), with the CS and US co-terminating at the same moment at the end. After repeated CS-US presentations with a constant ISI, the cerebellum generates a well-timed conditioned response (CR, i.e., eyelid closure) that anticipates the US. CEBC has long been considered a reference paradigm to investigate the neuronal underpinnings of sensorimotor adaptation and the alterations caused by cerebellar lesions [4–6]. The neural pathways involved in CEBC have been identified: the CS is presumably largely conveyed by the mossy fibres (*mfs*) through the pontine nuclei, while US is encoded by the inferior olive and transmitted to the cerebellum via the climbing fibres (*cfs*). Nonetheless, the neural mechanisms of CEBC are still debated.

A core hypothesis is that CEBC depends on suppression of simple spike (SS) activity in Purkinje cells (PCs) [7–9]. Inspired by the well-designed matrix architecture of the cerebellar cortex, Marr [10] was one of the first to propose a central role for parallel fibre (*pf*) to PC synapses in motor learning. Yet, it were Albus [11] and Ito [12], who specifically advocated a role of long-term depression (LTD) at the *pf*-PC synapse as the main mechanism underlying cerebellar learning. Although the *pf*-PC LTD mechanism has been supported by experimental evidence [2,13,14], recent results are challenging its dominant role in cerebellar learning. First, there are several other forms of plasticity distributed over multiple synaptic sites, some of which might also play a role in learning [14,15]. Second, in absence of any synaptic input, PCs exhibit high-frequency intrinsic firing, which presumably can hardly be reduced by a decrease in the activity of its direct *pf* excitatory input [16,17]. This suggests that *pf*-PC LTD by itself cannot be sufficient for the SS suppression of PCs during CEBC [18,19], and that synaptic inhibition [20–22] from molecular layer interneurons (MLIs) is likely to play a major role in driving PC-SS suppression [19,23,24]. Even so, recent experiments using double knock-out mice have shown that a complete blockage of CEBC requires not only MLI feedforward inhibition to be switched-off, but also a blockage of *pf*-PC LTD [25]. Given that the relevant MLIs are presumably driven by the same *pfs* that excite the PCs responsible for SS suppression, these

findings indicate that the feedforward inhibition exerted by the MLIs serves to bring the SS activity actually down, while it may be the main role of *pf*-PC LTD to prevent potentiation of the PCs [26].

Finally, while PC-SS suppression is critical for expression of the conditioned responses (CRs) of the eyeblinks [26], SS facilitation may also contribute, to some extent, to spatiotemporal control of the CRs [19,27]. This suggests that multiple cerebellar microzones and modules receiving at least in part different input signals may be involved in CEBC: downbound modules, including zebrin-negative (Z-) microzones, where PCs are more likely to undergo SS suppression, and upbound modules, potentially including zebrin-positive (Z+) microzones, where PCs are more likely to undergo SS facilitation. Whereas PCs in the downbound modules show prominent velocity-related responses, those in the adjacent upbound modules appear to show an increase in activity that is antagonistic to the response of the downbound PCs and is more proportional to the position of the eyelid during the CR [28]. Whether the downbound and upbound modules correspond perfectly or only *grosso-modo* to the zebrin-negative (Z-) and zebrin-positive (Z+) microzones, respectively, remains to be shown [26,28–30]. Thus, the differential properties in the firing patterns of PCs and deep cerebellar nuclei (DCN) cells, in *pf*-PC plasticity, and in feedforward inhibition from MLIs to PCs, are all likely to contribute to fine-tuning of the motor response (e.g., see [31]), but the precise contributions of the downbound and upbound PCs in optimizing the balancing act during CEBC remains to be elucidated.

Here we set out to model how a relatively simple associative paradigm like CEBC can be facilitated by the different modules of the olivocerebellar system. Computational models of the cerebellum can be effectively used to investigate cerebellar circuit functioning and to explain *in vitro* and *in vivo* experimental studies [32]. Starting from models with multicompartmental neurons and detailed synapses, simplified bioinspired Spiking Neural Networks (SNNs) have been developed and validated [33,34]. Once embedded in closed-loop control systems [35,36], cerebellar SNNs can drive autonomous learning in sensorimotor tasks and can be used to simulate the underlying neurophysiological mechanisms and to investigate the impact of alterations in pathological conditions [37,38]. Based on the SNN models of CEBC that have been developed in the last decades to address specific mechanistic questions [33–35,37,39–41], we have extended our realistic spiking model of the olivocerebellar circuit [42,43] to include multiple forms of synaptic plasticity [14,32] distributed across downbound and upbound microcomplexes [26]. Once embedded in a sensorimotor control system, the model was trained to learn CEBC. Model simulations predicted a synergistic role of plasticity at *pf*-MLI synapses and *pf*-PC LTD in regulating PC-SS firing and a prominent (but not exclusive) role of the downbound microcomplex in controlling CEBC, unifying a broad set of experimental observations. For example, the combined switch-off of MLI feedforward inhibition and *pf*-PC LTD let emerge the CEBC alterations observed in double knock-out mice [25]. Ultimately, our CEBC model can serve to further develop our unifying hypothesis of cerebellar function, adding elements on the underlying circuit operations that make the cerebellum a general-purpose co-processor [32,44,45].

## 2. Methods

Simulations in this work are based on experimental datasets recorded from control and mutant mice during CEBC (details below, 2.1). An olivocerebellar network model was reconstructed by wiring models of three regions (Fig 1A): i) the cerebellar cortex, ii) the DCN, and iii) the inferior olive (IO). This way we expanded processing in a microzone (a group of functionally related PCs and their inputs in the cerebellar cortex), to that in a microcomplex (a

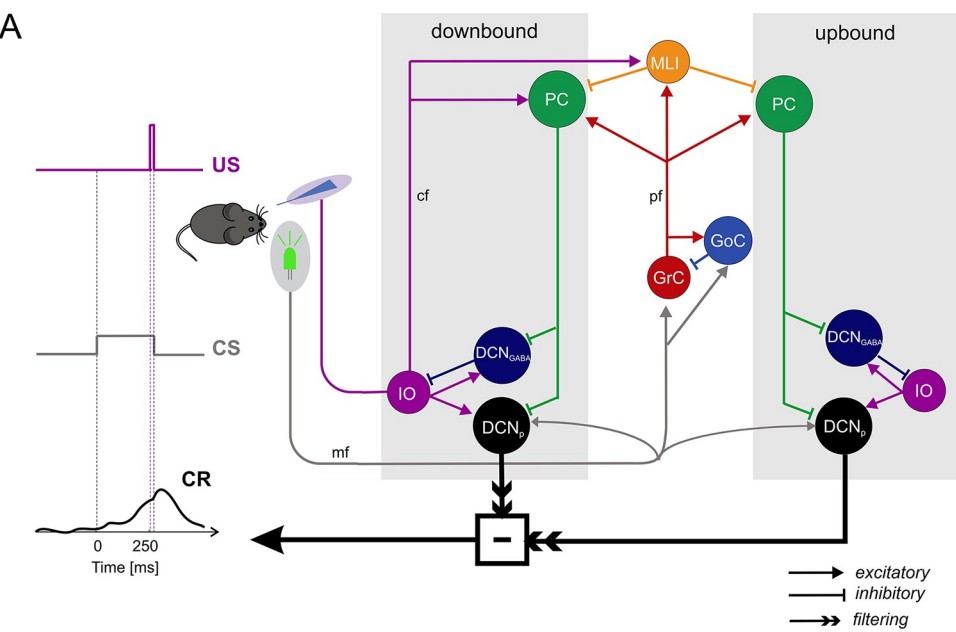

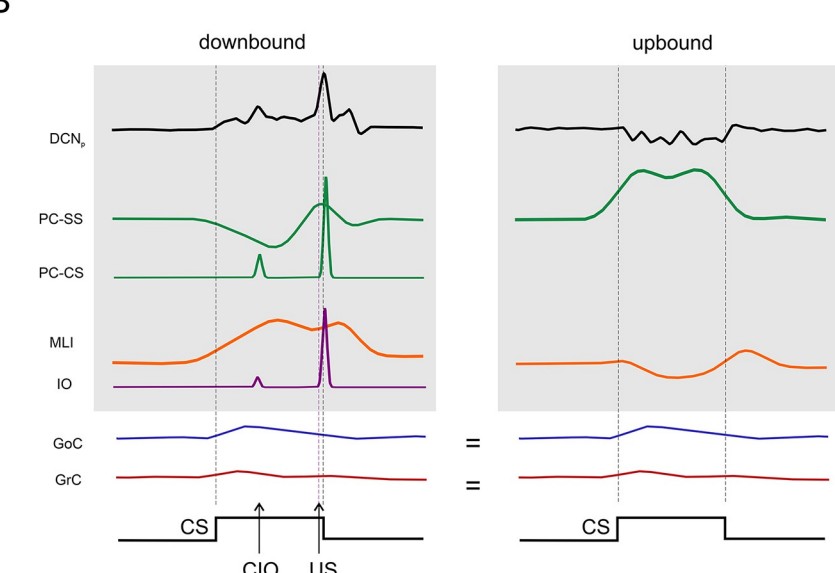

**Fig 1. Network model structure and dynamics.** (**A**) Schematics of the olivocerebellar circuit model made of two micromodules including a cerebellar cortical microzone along with the DCN and IO. The downbound and upbound microcomplexes differ for their *cf* input and for neuronal and plastic mechanisms, but share the same mossy fibre input and molecular layer interneurons. The CS and US input signals are conveyed through the *mf* and *cf* pathways, respectively, which activate the circuit determining the DCN output activity. The contribution of downbound and upbound $DCN_p$ neurons is convolved with a gaussian filter and algebraically summed to generate the behavioural output. The conditioned response, CR, is learned through plasticity mechanism and anticipates US when $DCN_p$ activity crosses a behavioural threshold. The mouse icon was adapted from https://openclipart.org/. (**B**) Sequence of signals (SDF) in the neuronal populations of the downbound and upbound micromodules (the sample traces are taken at the end of CEBC training). The CS conveyed through the *mfs* modulates the GrC, GoC and $DCN_p$ cell excitation. GrC activity propagates to PCs and MLIs. The PC-SS firing in turn inhibits the DCN cells, differentially modulating their activity in the downbound and upbound microcomplexes. The US, encoded in IO neurons and conveyed through the *cfs*, elicits a complex spike in PCs (PC-CS) and increases the activity in MLIs; note that the PC response is shown separately for simple and complex spikes. A conditioned IO response (CIO), which emerges during learning, causes a similar chain of events. Following *cfs* activation, inhibition from MLIs, in conjunction with intrinsic rebound properties of PCs and DCN cells, causes well-timed modulation in PCs and DCN cells. The arrows indicate the sequence of electrical events in the network. The network operates in feedforward mode, except for $DCN_{GABA}$ ➔ IO

feedback, which tends to reduce IO activity. Note that US signals forwarded by the *cfs* only reach the downbound module. *Acronyms*: US, unconditioned stimulus; CS, conditioned stimulus; CIO, conditioned IO response; CR, conditioned response; *mf*, mossy fiber; *cf*, climbing fiber; *pf*, parallel fiber; GrC, granule cell; GoC, Golgi cell; PC, Purkinje cell; PC-SS, Purkinje cell simple spike; PC-CS, Purkinje cell complex spike; $DCN_p$, externally-projecting deep cerebellar nuclei cell; $DCN_{GABA}$, GABAergic deep cerebellar nuclei cell; IO, inferior olivary cell.

microzone of PCs connected to their target cells in the DCN), and to that in a micromodule (a microcomplex connected to the neurons in the IO) [26]. The model was endowed with two long-term plasticity sites and differentiated microcomplexes, which were coupled to a motor output decoder to simulate CEBC. The model was first tuned to reproduce the experimental basal discharges (details below, 2.2) and activity modulation of the main neural populations in CEBC simulations (details below, 2.3). Specific lesions were then introduced in the network (details below, 2.4), so as to alter motor behaviour. The analysis of simulated neuronal activity and motor output (details below, 2.5) in control and lesioned conditions allowed to predict the neural underpinnings of CEBC in control and mutant mice [25] (S1 and S2 Figs).

## 2.1 Reference experimental datasets

The reference experimental dataset used for model construction and validation included multiscale data acquired during CEBC [19,25,27]. As described in detail in the corresponding studies, animals were trained throughout 10 daily learning sessions, each made up of 100 CS-US paired trials. The CS was a 260-ms green LED light, while the US was a 10-ms air puff co-terminating with the CS, resulting in an Inter-Stimulus Interval (ISI) of 250 ms, which is supposed to maximize CEBC performance. Eyelid closure was recorded [46] and used to compute the amplitude, timing and % of CRs across learning sessions.

The dataset included both the %CR learning curve and single-unit recordings from 28 PCs, 13 MLIs and 70 $DCN_p$ cells (DCN GAD-negative large cells projecting out of the cerebellum; see details below 2.2) acquired in C57Bl/6 mice (control) at the end of learning (after the last training session). Average spiking rates were computed for each trial, during a 500-ms window before the stimuli to quantify baseline frequency, and during the ISI to quantify firing changes (suppression or facilitation, and trained frequency) correlated with motor learning (Table 1). The analysis excluded the first 50 ms of the ISI, where only fast non-associative responses are supposed to occur.

Furthermore, complex spikes were extracted from PC *in vivo* recordings. This analysis showed that throughout learning, a complex spike emerged during the ISI (conditioned complex spike), with 43% probability of occurrence in fully trained mice, at an average latency of 88 ms after CS onset. To date, the mechanism causing this complex spike response (and thus IO/*cf* activity that is the source of complex spikes) remains unknown, but it may well originate from extra-cerebellar regions [47].

In addition, information on alterations in motor behaviour was derived from CEBC experiments in Knock-Out (KO) mice: i) *GluR2Δ7* mice (*pf*-PC LTD KO), ii) *L7-Δγ2* mice (MLI

**Table 1. Reference experimental neural data.**

| Cell type | Baseline frequency [Hz] | % firing change | Trained frequency [Hz] |
|---|---|---|---|
| PC (simple spikes) | 97 ± 28 | -14 ± 12 (suppression) | 83 ± 27 |
| MLI | 20 ± 11 | +321 ± 325 (facilitation) | 66 ± 39 |
| $DCN_p$ | 67 ± 27 | +88 ± 57 (facilitation) | 126 ± 47 |

Firing rates during baseline and their modifications at the end of learning, in the reference experimental datasets on control mice.

output KO), and iii) *GluR2Δ7-L7-Δγ2* mice (double KO). Specifically, CEBC was not significantly compromised in i), and it was only slightly affected in ii) [19]. Double KO mice showed almost no CEBC learning, demonstrating that the two plasticity mechanisms synergistically cooperate to induce CEBC [25].

## 2.2 Cerebellar network model and tuning steps

The present model (Fig 1A) was built to reproduce realistic properties of neurons, connectivity, modularity, and plasticity rules. Single neuron dynamics (see Par. 2.2.2) allow a faithful encoding of input signals, like the US. The burst-pause mechanisms that are triggered in the IO-PC-DCN loop are among the most faithful representations of the events associated with climbing fibre signalling that have been achieved with a spiking model (cf. [33,48]). Connectivity derives from a study based on neuronal morphology and intersection rules generating precise structural and functional relationships among the circuit elements [42]. Multiple and differentiated modules for downbound and upbound microzones are introduced for the first time here (although the concept was anticipated in [35]). Plasticity in the MLI inhibitory networks is introduced here for the first time and combined with *pf*-PC bidirectional plasticity, which was already embedded in previous models [33–35,37,39–41].

**2.2.1 Network architecture.** The olivocerebellar network (Fig 1A) was reconstructed using the Brain Scaffold Builder framework (BSB; RRID:SCR_008394, version 3.8+) [42]. The cerebellar cortex model was derived from a recent detailed reconstruction [42] including both the granular and molecular layer. The DCN and IO models were derived from previous reconstructions [43, 49] and scaled proportionately with respect to the cerebellar cortical model. Neurons were placed in the network volume based on their morphological features and volumetric densities taken from literature, and in/out degree statistical rules were applied for connectivity. The entire olivocerebellar network was made of point-neuron models.

i. The cerebellar cortex model corresponded to a volume of 17.7 $10^{-3}$ mm$^3$, contained 29'230 neurons (Granule Cells, *GrC* with their axons–the ascending axon *aa* and the parallel fiber *pf*; Golgi Cells, *GoC*; Purkinje Cells, *PC*; Molecular Layer Interneurons, *MLI*, namely Stellate Cells, *SC*, and Basket Cells, *BC*) plus 2'453 other elements (mossy fibers, *mf*; glomeruli, *glom*).

ii. The DCN model contained two main neural populations: i) GAD-negative large, which are glutamatergic neurons projecting outside the cerebellum (DCN$_p$); ii) DCN$_{GABA}$, which are GABAergic neurons that inhibit IO neurons and form the olivocerebellar loop [50].

iii. The IO model contained neurons of one cell type only [51].

Recent studies suggested that two different micromodules are involved in CEBC [26]. Therefore, PC, DCN and IO neuronal populations in our network model were part of and connected to two adjacent microzones based on their position in the parasagittal direction: i) a downbound microzone (with Z- PCs) containing 70% of PCs and connected to 70% of nuclear and olivary cells, and ii) an upbound microzone (with Z+ PCs) for the other 30%. These account for 2/3 of PCs with a higher baseline firing rate (more likely to undergo simple spike suppression during conditioning) and 1/3 of PCs with a lower baseline firing rate (more likely to undergo facilitation), respectively, as in experimental recordings [17,26,52].

In the cerebellar cortex model, input signals from *mf* are transmitted to the granular layer via excitatory connections through the glomeruli (*mf*-glom, glom-GrC, glom-GoC). The granular layer includes recurrent excitatory (from GrC) and inhibitory (from GoC) connections (GoC-GrC, GoC-GoC, GrC(*aa*)-GoC, GrC(*pf*)-GoC). Signals are then transmitted to Purkinje

and Molecular layers through *aa* and *pf* projections (GrC(*aa*)-PC, GrC(*pf*)-PC, GrC(*pf*)-SC, GrC(*pf*)-BC). In turn, MLIs inhibit both PCs and other MLIs (SC-PC, BC-PC, SC-SC, BC-BC connections). In the DCN, $DCN_p$ neurons receive excitation from *mf* (*mf*-$DCN_p$) and IO (IO-$DCN_p$), and inhibition from PCs (PC-$DCN_p$), while $DCN_{GABA}$ neurons are inhibited by PCs (PC-$DCN_{GABA}$) and excited by IO collaterals (IO-$DCN_{GABA}$). IO neurons transmit excitatory signals to PCs, causing complex spikes (IO-PC), and to MLIs through spillover connections (IO-SC and IO-BC). IO neurons also receive feedback inhibition from DCN through $DCN_{GABA}$-IO connections [53]. In our model, all connections involving PC, DCN and IO neurons are confined within a particular micromodule. Moreover, MLIs, the axons of which traverse predominantly in the sagittal plane [26], are also associated with upbound and downbound microzones based on whether they inhibit more upbound or downbound PCs, respectively. In summary, the cerebellar cortex model contains 15 connection types [42], and the extension to the deep cerebellar and olivary nuclei brought out 9 more connection types. The detailed network architecture information is reported in Table A for neurons and Table B for connectivity in S1 Text).

**2.2.2 Neuron and synapse models.**   The network was simulated as a spiking neural network made of point neurons, modelled as Extended-Generalized Leaky Integrate and Fire (E-GLIF). This model allows to keep the main electroresponsive features of cerebellar neurons, while reducing the computational load of simulations [54], making it feasible to simulate a large-scale SNN with realistic population sizes and millions of plastic synapses, even if requiring high-performance computing resources (see Par. 2.4). Population-specific parameters were used [55] to reproduce the different spiking patterns of cerebellar neurons identified in experimental recordings (Table A in S1 text). For PCs, the parameter controlling spontaneous firing was set to different values for Z+ PCs and for Z- PCs using a linear fit of the current-frequency relationship to match their basal discharge *in vivo* (i.e., 176.3 nA for Z+ PC corresponding to 39±1 Hz and 742.54 nA for Z- PCs corresponding to 86±1 Hz discharge). Glomeruli and mossy fibers were modelled as relay units, transmitting the same spike trains that they receive.

Synaptic inputs were modelled as alpha-shaped conductance-based synapses, with delays and decay rates taken from literature experimental data and specific for each connection type [43]. The number of synaptic contacts was taken into account by creating multiple connections per pair. Synaptic weights, i.e., the synaptic conductance changes induced by presynaptic spikes, were tuned through bisection search initialized with the values in [43], to obtain the physiological basal discharges in each neuronal population as recorded in behaving mice [19,22,27,56,57].

The connectivity and synapse parameter values are reported in Table B in S1 Text.

**2.2.3 Plasticity models.**   In order to simulate learning neural mechanisms [58], the network model included long-term plasticity at the *pf*-PC and *pf*-MLI synapses. The learning rules implemented bidirectional spike-driven plasticity supervised by the *cf*s.

The *pf*-PC plasticity rule was derived from previous works [34,36], based on the observation that *pf* stimulation coupled with *cf* activation (teaching signal) triggers Long-Term Depression (LTD) at synapses between *pf*s and PCs receiving the *cf* signal, while *pf* stimulation alone causes *pf*-PC Long-term Potentiation (LTP) [13] (see also Equation A in S1 Text).

The *pf*-MLI plasticity rule was defined *de-novo* and implemented in NEST. The *pf*-MLI plasticity was constructed following the same principle of *pf*-PC plasticity, i.e., using a mechanism based on co-activation of *pf*s and *cf*s, but the sign of changes was reversed: a *cf* spike (teaching signal) causes LTP at *pf*-MLI synapses between *pf*s that are active just before *cf* activation and MLIs receiving the *cf* signal. The LTP amount depends on the convolution of *pf* spikes with a kernel function. The kernel function for *pf*-MLI LTP was modelled as an alpha function,

based on experimental data on MLI activity modulation occurring during learning [19]. The rise time τ of the alpha-function was set to 50 ms, to account for the peak time in MLI activity modulation within the inter-stimulus interval (S3A Fig) [19]. Otherwise, if a *pf* fires without any coincident *cf* activation, the corresponding synapse undergoes LTD. The rule was implemented as follows:

$$\Delta W_{pf_i \to MLI_j}(t) = \begin{cases} LTP_{MLI} \int_{-\infty}^{t_{cf\_spike_j}} K(t_{cf\_spike_j} - x)\delta_{pf_i}(t_{cf\_spike_j} - x)dx, \text{ if } pf_i \text{ is active and } t = t_{cf\_spike_j} \\ \\ -LTD_{MLI}, \text{ if } pf_i \text{ is active and } t \neq t_{cf\_spike_j} \\ \\ 0, \text{ otherwise} \end{cases}$$

Where:

$$\delta_{pf_i}(t) = \begin{cases} 1, \text{ if } pf_i \text{ is active at time } t \\ 0, \text{ otherwise} \end{cases}$$

And the Kernel function is:

$$K(t) = \frac{t}{\tau} \cdot (e^{1-\frac{t}{\tau}}) \cdot \chi_{[t0,+\infty)}(t)$$

Where:

$$\chi_{[t0,+\infty)}(t) = \begin{cases} 1, t \in [t0, +\infty) \\ 0, \text{ otherwise} \end{cases}$$

The learning rates at each plastic site, i.e., $LTP_{PC}$, $LTD_{PC}$, $LTP_{MLI}$, $LTD_{MLI}$, were free parameters that required tuning. They were tuned using trial-and-error in order to maximize the match between the experimental and the simulated firing modulation at the end of a short CEBC sequence (20 trials), for MLI, PC and $DCN_p$ populations in the downbound microzone. Once the balance between these parameters was fixed, the final parameter values were derived applying a linear scaling to account for longer sequences (×50) like those recorded experimentally.

The firing rate was computed as Spike Density Function (SDF) from spike trains and the firing modulation was computed applying the same algorithm as in experimental data processing [19] (details in section 2.5).

In summary, the network underwent a multi-step tuning procedure, using experimental data from control mice performing CEBC. Specifically, first, the initial synaptic strengths (weights) of all the connection types were set by maximizing the match with the experimental baseline discharges of each neural population. Secondly, the plasticity rates for the plastic connection types were tuned using the firing modulation measured in control mice at the end of learning as a target.

## 2.3 CEBC protocol

To simulate CEBC, the following CS and US inputs were used:

- CS: a 10 Hz Poisson spike pattern lasting 260 ms was delivered to the *mfs* in a cylinder with ellipsoidal basis centred in the granular layer and containing 45 out of 117 *mfs*. This allowed

to avoid border effects. The granular layer, with this asset, performed a non-recurrent sparse encoding of time intervals [59].

- US: a 500-Hz burst lasting 10 ms and co-terminating with CS was delivered to IO neurons in the downbound micromodule [19].

- Conditioned IO response (CIO): a 400-Hz burst was delivered to half of IO neurons in the downbound microcomplex, with latency 88 ms after CS onset [19], causing a conditioned complex spike in PCs through *cf*. The occurrence probability of this input was linearly increased with the number of trials raising up to 43% at the end of learning. Given that the mechanistic origin of the CIO is still unknown and could potentially be extra-cerebellar [47], this input in the model was set based on experimental observations, and the parameters, e.g. probability of occurrence and latency, were extracted from Ten Brinke and colleagues [19]. The effect of the CIO on the whole network activity and behaviour was unconstrained and followed the mechanisms embedded in the SNN.

Each trial included an initial 500-ms baseline window with a background 1 Hz random input to all *mfs*, consistent with observations of low-frequency firing of *mfs in vivo* at rest [60]. Simulations included 10 blocks of 100 CS-US trials [19].

The output motor response (eyelid closure) was driven by the $DCN_p$ population. The spike patterns of the $DCN_p$ neurons of both microcomplexes were decoded into a net analog signal representing the cerebellar motor output. The decoding strategy was a rate-based conversion applying a 20-ms moving average filter on the $DCN_p$ spike trains convolved with a Gaussian kernel window (25 ms width). These processing steps were introduced to account for smoothing and delay in the efferent sensorimotor system from the cerebellum to the periphery [35,61]. The net signal between downbound and upbound $DCN_p$ was computed and then normalized between 0 and 1, subtracting the baseline in each trial and dividing by the maximum amplitude (i.e., maximum eye closure) of the control simulations.

A CR was generated if the motor output reached a threshold in the last 200 ms of the ISI (CR window). Similar to experimental data, the threshold determining the CR onset in each trial was set to:

$$threshold = mean(baseline) + 2.5*STD(baseline)$$

This accounts both for baseline mean value and variability through its Standard Deviation (STD). Only threshold values higher than 0.2 were allowed. The time instant of threshold crossing was considered as the CR onset latency, while the time of CR peak was considered as the CR peak latency.

## 2.4 CEBC simulations with control and lesioned cerebellar network models

All simulations were run in parallel on 36 cores through MPI on the CSCS *Piz Daint* supercomputer, with a time resolution of 1 ms, using NEST 2.18 [62], through the Python interface [63]. Cerebellum-specific neuron and synaptic plasticity models (Par. 2.2.2 and 2.2.3) were implemented as a NEST extension module, *cereb-nest*, available at https://github.com/dbbs-lab/cereb-nest.

Once all the free parameters were set in control condition, the complete CEBC protocol was carried out.

Then, the network was modified to emulate the mutant mice, i) *GluR2Δ7*, ii) *L7- Δγ2*, iii) *GluR2Δ7-L7-Δγ2*. The first pathological model was created by switching off *pf*-PC LTD

($LTD_{PC} = 0$). The second one was created by disconnecting MLIs from PCs (SC-PC and BC-PC weights = 0). The third network model included both alterations.

For each model (control and three KO networks), 3 simulations of CEBC protocol were carried out, each one with different random seeds for the *mf* background and single neurons noise.

## 2.5 Data analysis

Given the multiscale nature of the model, the microscale mechanisms underlying associative learning were analysed, specifically the simulated spiking activity of the main neural populations and the connection weights at plasticity sites.

Population baseline frequencies were computed as number of spikes during the 500-ms baseline window of the first trial in each simulation, divided by the number of cells in the population.

The spiking activity of PCs was analysed by generating two time series to separate PC simple spikes (PC-SS) from complex spikes (PC-CS). PC-CS corresponded to bursts with a frequency that was 2-times higher than the baseline firing frequency. PC-SS were all the other spikes (when a PC-CS occurred, it was substituted with a single spike in the PC-SS time series).

The spiking activity of MLI, PC simple spikes (PC-SS) and $DCN_p$ neurons was evaluated computing the SDF, which was obtained by convolving spike times with a Gaussian kernel (41-ms width for MLI and PC-SS, 10-ms width for $DCN_p$) in each trace (i.e., each cell in each trial for all simulations) (S2 Fig). The same was applied to PC-CS and IO spikes, with a 5-ms Gaussian kernel.

To assess the functional validity of the model, neural activity modulation (suppression and facilitation) at the end of learning was computed, analogously to experimental data analysis [19]. Specifically, for each SDF trace in the last block, we computed the average % deviation in the CR window with respect to the average baseline $SDF_{baseline}$, separating time instants with suppression and with facilitation:

$$suppression = \frac{\sum_{i=1}^{Ns} \frac{SDF_i}{SDF_{baseline}} - 1}{2}\%$$

for the *Ns* time instants *i* where $SDF_i < SDF_{baseline}$

$$facilitation = \frac{\sum_{i=1}^{Nf} \frac{SDF_i}{SDF_{baseline}} - 1}{2}\%$$

for *Nf* time instants *i* where $SDF_i > SDF_{baseline}$

Traces with significant modulation at the end of learning were considered, identified as the ones where SDF modulation in the CR window was higher than the deviation during baseline.

PC and $DCN_p$ populations were analysed separately for the downbound and upbound microcomplexes, where they are supposed to undergo opposite modulations of activity. Instead, the MLIs were considered all together, distinguishing downbound-preferring and upbound-preferring, depending on the specific neural activity modulation (facilitating and suppressing, respectively) [26].

SDF signals were reported normalized with respect to average baseline of each trace ($SDF/SDF_{baseline}$), as in the experimental studies, except for PC-CS and IO spikes in the downbound microcomplex, for which $SDF_{baseline}$ is ~0.

Finally, to quantify the underlying synaptic modifications, for each plasticity site, we represented plasticity curves, computing in each block the average % weight change with respect to

the initial weight, multiplied by the relative number of synapses undergoing such change (LTD or LTP).

## 3. Results

The olivocerebellar network model, including downbound and upbound microcomplexes [26], differentially received US and CS at the input and regulated CR at the output (Fig 1A). Compared to the upbound, the downbound microcomplex was larger (70% of the PCs and DCN neurons), had PCs with higher basal firing frequency, and received the US from corresponding IO neurons. After tuning the network activity and obtaining behavioural responses under physiological conditions (see Methods and S1 Fig), CEBC simulations matched the physiological parameter range measured in mice [19,25,27,64,65] concerning basal firing rate and firing rate modulation after learning (Table 1 and S3B and S6 Figs), and CR learning rate and timing in control conditions (see below Par. 3.2).

In the model, the CS was conveyed through the *mfs*, while the US and the CIO response, encoded in downbound IO neurons, were conveyed through the *cfs*. Simulations revealed a cascade of synaptic events summarising current physiological knowledge about the olivo-cerebellar circuits (exemplified in Fig 1B for the *downbound* and upbound *microcomplex*). The *mfs* increased GrC, GoC and $DCN_p$ cell excitation. GrC activity propagated to PCs and MLIs. When PC-SS firing increased, this in turn inhibited DCN cells. IO neurons, through the *cfs*, elicited PC-CS and increased MLI activity. Following *cfs* activation, inhibition from MLIs modulated PCs and, indirectly, DCN cells, while the intrinsic rebound properties of PCs and DCN cells were in place. The network operated entirely in the feedforward mode except for $DCN_{GABA}$ ➔ IO feedback, which tended to reduce IO activity (see also Discussion). Downbound $DCN_p$ neurons showed two evident pauses in the response profile. The first pause, caused by the PC-SS driven by *mf*-GrC-PC excitation, inhibited $DCN_p$. The second pause, caused by the conditioned IO response generating PC complex spikes, also inhibited $DCN_p$. It is worth noticing that these pauses are followed by rebound bursts caused by intrinsic DCN mechanisms and that the rebound bursts are reinforced by well-timed pauses in PCs. As explained below, these effects were modified by plasticity. Therefore, the cerebellar motor output could be tracked back to the activity of neurons and synapses providing a mechanistic explanation of the neuronal underpinnings of CEBC.

### 3.1 Microcircuit dynamics in downbound and upbound microcomplexes during learning

During learning, the firing dynamics of cerebellar cortical and nuclear neuronal populations changed due to synaptic plasticity driven by the US-related teaching signal conveyed by *cfs*. The direction of modulation (suppression or facilitation) for each neuronal population depended on the input signals received by the microzones: both CS and US in the downbound microzones, only CS in the upbound microzone. As shown in Fig 1A, GrCs, GoCs and MLIs were not differentiated between the two microzones (see Methods for details). The firing changes of the different neuronal populations along learning blocks are reported in Figs 2 and S5.

*MLIs*. Most SCs and BCs increased their firing rate during the ISI due to *pf*-MLI LTP. However, some MLIs decreased their firing rate during the ISI due to *pf*-MLI LTD (S6 Fig). The difference was due to the different probability of innervation by *cf* and *pfs*, with a gradient that favours *cfs* in proximity of the downbound microzone, effectively selecting downbound-preferring and upbound-preferring MLIs.

*Downbound PCs and $DCN_p$ cells*. PCs in the downbound microcomplex mainly underwent simple spike suppression generating a well-timed negative peak in the SDF profile before US.

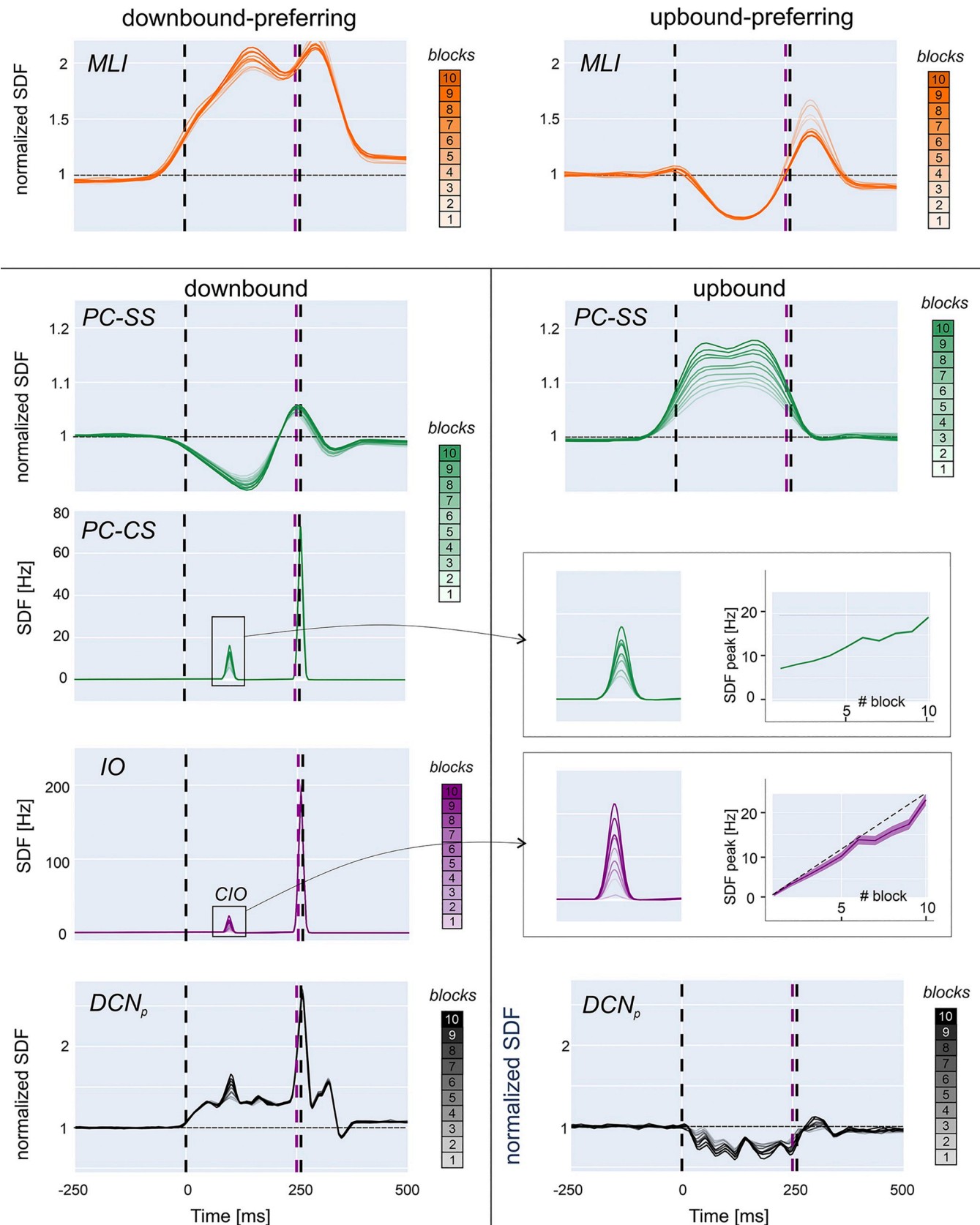

**Fig 2. Neural activity changes during learning.** Activity (SDF) in neuronal populations of the network along learning blocks (from light to dark lines). *Downbound-preferring MLIs*: the SDF progressively increases in the ISI across blocks due to the prevalence of *pf*-MLI LTP triggered by the US-related IO signal, with a peak before US onset. *Upbound-preferring MLIs*: the SDF progressively decreases in the ISI across blocks due to the prevalence of *pf*-MLI LTD triggered by *pf* spikes uncorrelated with IO activity. *Downbound*: PCs show SS suppression due to *pf*-PC LTD driven by the US-related IO signal and to downbound-preferring MLIs causing feedforward inhibition. PC-SS suppression is maximal before the US onset. DCN$_p$ cells activity increases due to reduced PC inhibition. The PC complex spike activity (PC-CS) increases due to the CIO, whose probability increases during learning. The inset shows that the peak of CIO activity (mean ± standard error across traces in each block) is lower than expected (dashed line) due to DCN$_{GABA}$ inhibition. *Upbound*: PCs show facilitation of SS activity due to *pf*-PC LTP and to upbound-preferring MLIs (a minority, cf Par. 3.3 results on neuronal recruitment) reducing their feedforward inhibition. DCN$_p$ cells show an SDF decrease due to enhanced PC inhibition. In all panels, SDF traces are normalized to baseline, and vertical lines indicate CS and US (cf. Fig 1). Traces are averaged across all cells and trials in each one of the 100-trial blocks.

Simple spike suppression of downbound PCs was caused by time-locked *pf*-PC LTD and *pf*-MLI LTP driven by *cf* signals. DCN$_p$ neurons in the downbound microcomplex increased their activity within the ISI during learning.

*Upbound PCs and DCN$_p$ cells.* PCs in the upbound microcomplex mainly underwent simple spike facilitation all along the ISI. Simple spike facilitation of upbound PCs was caused by *pf*-MLI LTD and *pf*-PC LTP driven by *pfs* spikes, since the upbound microcomplex received only the CS signal (cf. Fig 1A and see Methods 2.2.3). Consequently, upbound DCN$_p$ neurons mainly decreased their firing because of increased PC inhibition. They also showed activity oscillations consistent with experimental findings [64]. Therefore, upbound DCN$_p$ suppression contributed to the net motor output by setting its amplitude range (see below and S6 Fig).

In the downbound microcomplex, a PC complex spike during the ISI emerged during learning due to the CIO (Fig 2). This caused rebound bursting in DCN$_p$ cells, amplifying the CR. When the CR was produced, the feedback loop from DCNGABA to IO limited the CIO increase (cf. Fig 1).

In aggregate, simple spike modulation of PC and DCN$_p$ cells peaked during the ISI and complex spike modulation evolved as observed in CEBC studies in mice [19,27], providing an important element for model validation.

## 3.2 Simulated CEBC in control conditions and with altered plasticity

The net signal obtained from downbound and upbound DCN$_p$ cells was used to generate the motor output, i.e., eyelid closure. The DCN$_p$ signal progressively increased and anticipated its rise in the ISI, i.e., before US. The DCN$_p$ responses in the two microcomplexes followed a different time evolution, so that the combined contribution of both microcomplexes was needed

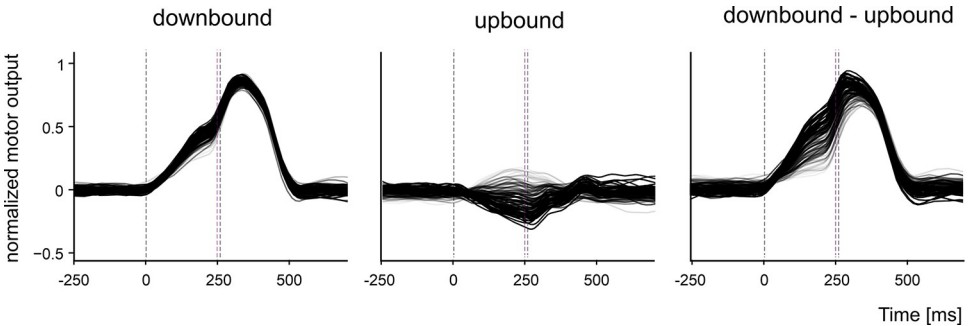

**Fig 3. Contribution of the downbound and upbound microcomplexes during learning.** The motor output is decoded in three conditions during learning: only from downbound DCN$_p$ activity, only upbound DCN$_p$ activity, from both downbound and upbound DCN$_p$. Beside the major contribution of the downbound microcomplex, it should also be noted the sizeable contribution of the upbound module in shaping the compound response. Each motor output trace is the average of 10 contiguous trials.

to shape the timing and amplitude of CR. Eventually, the $DCN_p$ signals were convolved with an output function yielding a sequence of behavioural responses (Fig 3), which were subsequently thresholded to obtain the CEBC learning curve.

In control, the simulated CEBC learning curve attained 81% CR (last block) following a bounded exponential curve, closely matching the experimental one measured in mice [19] (Fig 4A). Similar motor output and learning curves were obtained for simulations with shorter (200 ms) and longer (300 ms) ISI durations, proving the robustness of the model in fine-tuning the timing of the motor output (S4 Fig). The SNN was then modified to mimic the circuit alterations following two specific genetic mutations: *GluR2Δ7* KO that eliminates *pf*-PC LTD, and *L7-Δγ2* KO that eliminates the MLI output [25] (S1 Fig). In simulations of *pf*-PC LTD KO, the learning speed was lower but eventually CR approached control values (73%) in the last learning block. In simulations of MLI output KO, learning was almost unaffected all along the learning curve and attained 72% CRs in the last learning block. However, a severe CEBC impairment, with slower learning and just 41% CRs in the last block, was observed in the double KO. The absence of major CEBC deficits with individual mutations and a severe impairment with the double mutation closely matched the outcome of experiments in mice [25] (Fig 4B). Simulations also showed that CRs had significantly anticipated peak timing in MLI output KO, and significantly delayed onset and peak timing in *pf*-PC LTD KO and double KO (two-sample Kolmogorov-Smirnov test with 5% significance) (Fig 4C).

## 3.3 Predictions on microcircuit mechanisms with alterations in cerebellar cortex plasticity

Since the model was able to reproduce the behavioral impact of specific mutations, it was further used to unveil the underlying neural network changes that were not reported experimentally [25]. Akin with the concept that network learning is distributed and probabilistic in nature, the number of cells that showed significant changes during learning provided a close correlate of network alteration in mutant mice (Fig 5A). The discriminating factor explaining the CEBC alteration was the contribution of PCs and $DCN_p$ cells in the downbound microcomplex. For both cell types, the rank was: control > *pf*-PC LTD KO ~ MLI output KO > double KO and the changes strongly correlated with CEBC (Fig 5B). Upbound PC, $DCN_p$ and MLI responses did not change remarkably (<2%). Indeed, the number of cells showing significant learning parameter changes impacted on the behavioural response (Fig 5C). Another critical factor in determining network learning is PC-SS time-precision. A good proxy of PC-SS time-precision was the peak of SS suppression in the ISI (measured as probability distribution of minimum SDF times). In PCs, *pf*-PC LTD KO but much less so MLI inhibition KO reduced control over PC-SS timing (the cumulative square distance from control was 0.004 in *pf*-PC LTD KO, 0.002 in MLI output KO and 0.005 in double KO) (Fig 5D).

Further insight into the mechanisms of change was gained by analysing SDF modulation maps at the end of learning in the downbound microcomplex (Fig 6). In KO simulations, various alterations emerged in PC and $DCN_p$ response patterns, dominated by enhanced PC-SS firing and reduced $DCN_p$ cell firing in the ISI.

In control, most PCs were inhibited by MLIs, whose efficiency increased because of plastic changes at the *pf*-MLI synapse (see below). The reduced PC discharge together with the CIO and the *mf* input impinging onto $DCN_p$ cells raised the activity and changed the sequences of bursts, pauses and rebounds in $DCN_p$ neurons (cf. Fig 1B). The integration of these effects determined a time-locked increase in $DCN_p$ firing releasing behaviour, i.e., CEBC.

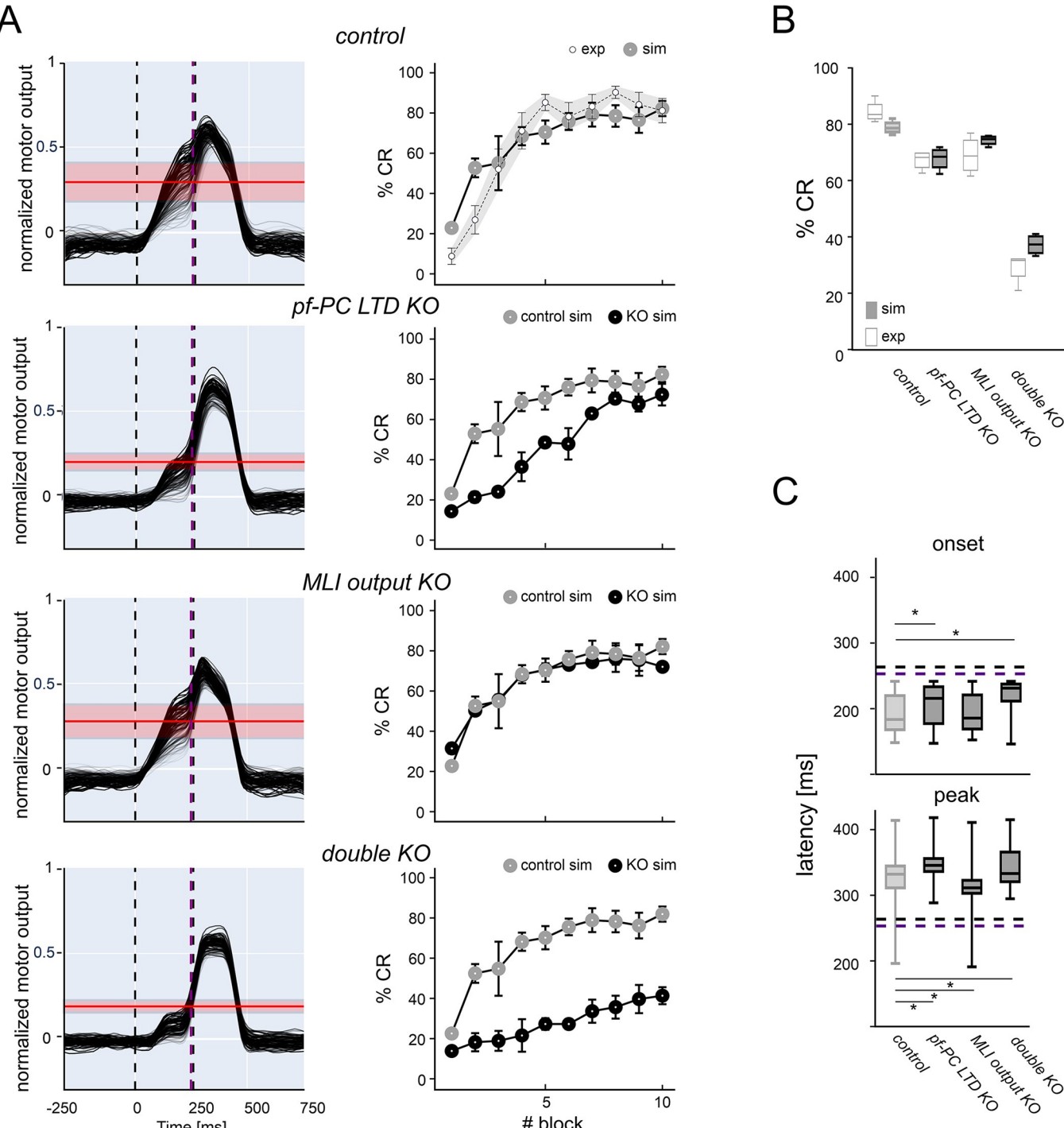

**Fig 4. Learning curves in control and KO simulations. (A)** The 4 rows show (from top to bottom) simulations in control, *pf*-PC LTD KO, MLI output KO, double KO. *Left*: the motor output decoded from DCN$_P$ activity (each trace is the average of 10 contiguous trials) shows a progressive increase of the response before the US as learning proceeds (light to dark lines). The mean and standard deviation of CR threshold are shown in red. *Right*: the graphs show average % CR in blocks of 100-trials. The %CR shows a progressive increase during learning. Bars indicate the standard deviation across trials and simulations. The top graph shows the control learning curve compared to experimental data. Each other graph shows the learning curve for the specific KO condition along with the control learning curve taken from top. **(B)** CR gain. The box-and-whiskers plots show %CR changes at the end of learning (last 3 blocks) for all the simulated conditions compared to experimental data. **(C)** CR timing. The box-and-whiskers plots show CR onset and peak latency (reported as time after the CS) for all the simulated conditions. Horizontal purple and black lines represent US onset and end, respectively. Asterisks mark significant differences.

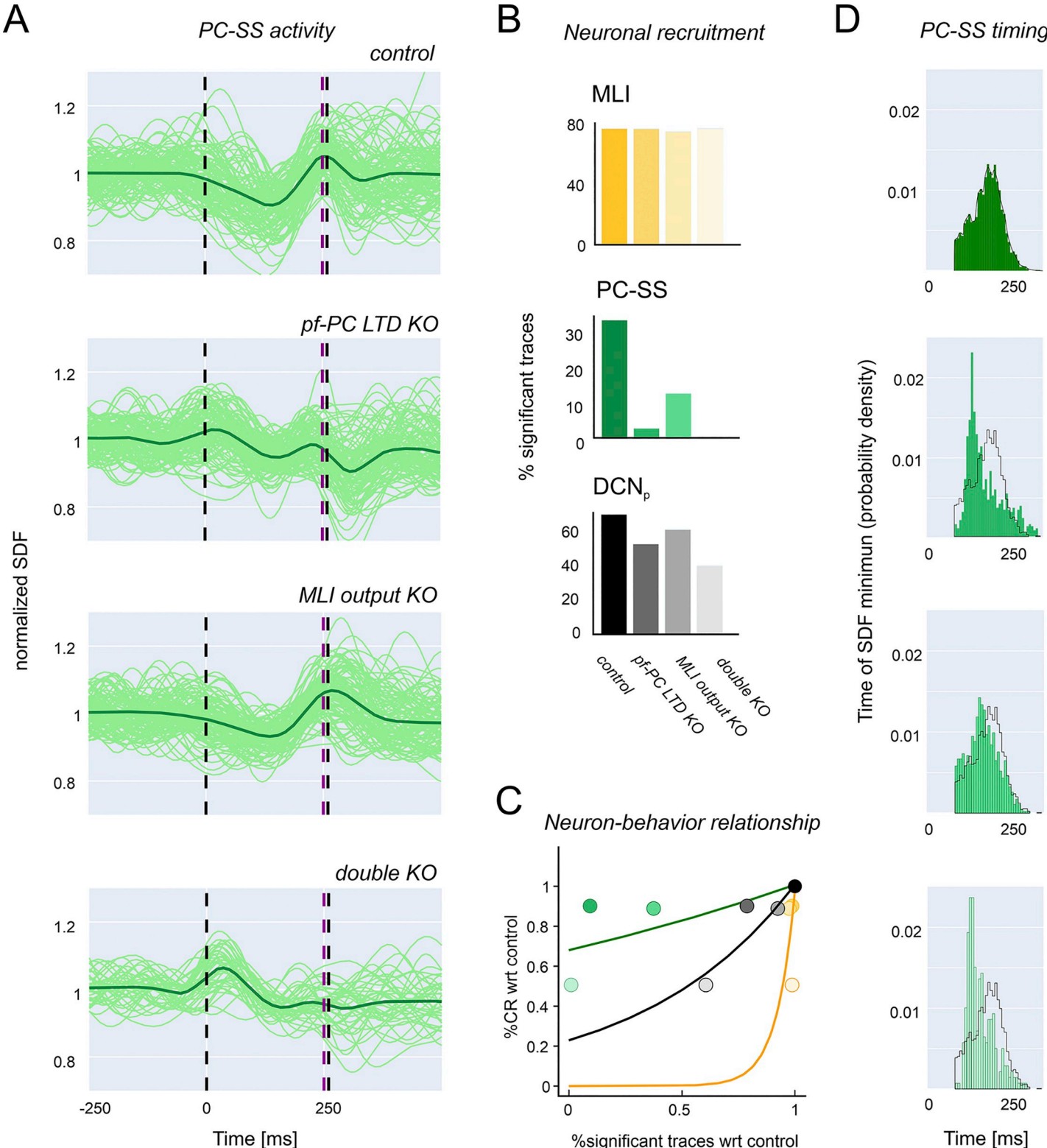

**Fig 5. Neural activity during learning in control and KO simulations. (A)** Discharge modulation in downbound PCs at the end of learning. SDF traces undergoing significant suppression or facilitation are shown normalized to baseline for the last simulation block along with the average trace (thick line). The traces are shown in control, *pf*-PC LTD KO, MLI output KO, double KO conditions. It should be noted that that *pf*-PC LTD KO reduces control over PS-SS synchronization (time-precision), while MLI inhibition KO reduces control over PC-SS average response amplitude (gain). **(B)** Bar graphs of the neuronal recruitment at the end of learning

for MLIs, PCs and DCN$_p$ cells in downbound microcomplex, both in control and KO simulations. Neuronal recruitment is measured as the % of traces undergoing significant suppression/facilitation in the last block of all simulations. **(C)** The plot shows the relationship between the behavioural response (%CR, from Fig 3B) and neural activity (%significant traces, from panel B) normalized to control. Note the steep non-linear correlation for both MLIs, PCs and DCN$_p$ cells in the downbound microcomplex (exponential fit), proving the prominent involvement of downbound neuron dynamics in controlling the behavioural output. **(D)** Probability distribution of SDF minimum times. The envelop of the control distribution is overlayed on the other histograms. Note the stronger difference with *pf*-PC LTD KO than MLI output KO.

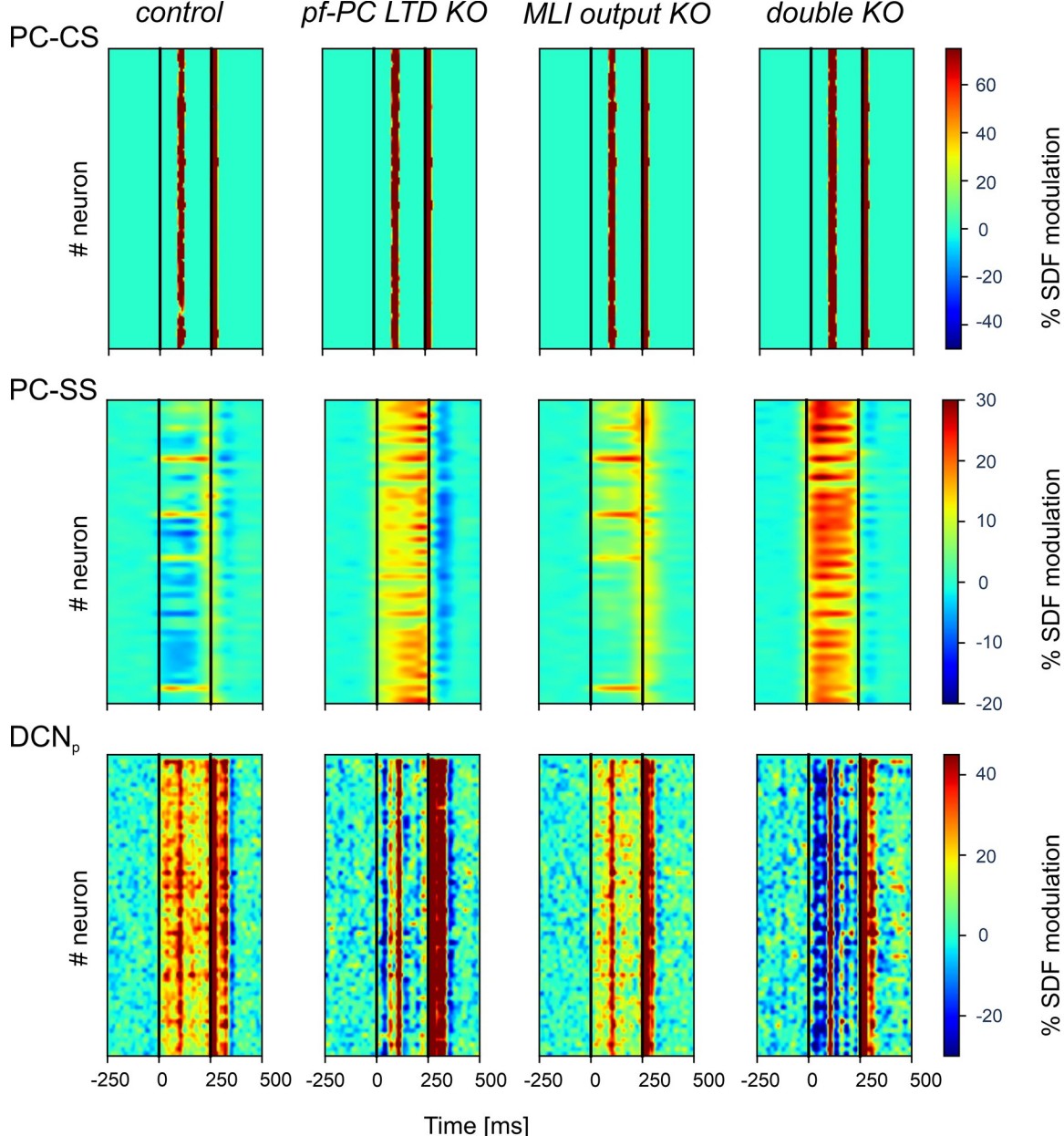

**Fig 6. Single neuron activity modulation at the end of learning in control and KO simulations.** The heat-maps show %SDF modulation for each of the PC and DCN$_p$ neurons of the downbound microcomplex of the last block with respect to pre-learning condition (first block) values (data are averaged across trials of the block). PC-CSs in the ISI are similar in all conditions. PC-SSs how an overall decrease in control but an increase in all KO conditions. DCN$_p$ neurons show changes opposite to the PC-SSs, due to the inhibitory effect of PCs on DCN. The residual facilitation in DCN$_p$ neurons largely reflects the PC-CS response in the inter-stimulus interval caused by the CIO. Note the much more severe and consistent alterations in the double KO with respect to each individual KO.

In *pf*-PC LTD KO, PCs were hyperactivated by granule cells, but received increased inhibition from MLIs thanks to *pf*-MLI plasticity, albeit not synchronized (S7 Fig). Together with bursting caused by CIO, this was sufficient to compensate for PC hyperactivation, returning some CEBC compensation at the end of learning (Fig 3).

In MLI output KO, PC inhibition was absent, but *pf*-PC LTD was sufficient to generate a moderate time-locked suppression of PC activity. This, combined with the *mf* drive, caused a moderate increase in $DCN_p$ cells activity, that could compensate for the lack of feedforward inhibition, returning some control on the CEBC (Fig 3).

In the double KO, the two alterations combined. The PCs discharged quite strongly because of intense *pf* transmission that was neither controlled by *pf*-PC LTD nor by MLI inhibition. Eventually, $DCN_p$ cells were strongly inhibited, but the sequence of burst, pause and rebounds triggered by CIO was sufficient to partially increase $DCN_p$ activity causing just a modest compensation.

In summary, our simulations of KO conditions suggest that *pf*-PC LTD fine-tunes the timing of CRs, providing a well synchronized PC-SS suppression, while MLI inhibition controls the amplitude of CRs, increasing the amount of PC-SS suppression (S7 Fig). Both are fundamental for proper CEBC learning and can partially compensate each other. In case these cortical mechanisms are damaged, the responses caused by CIO can explain the residual CEBC learning observed in experiments.

Simulations also predicted the redistribution of plasticity in KO conditions (Fig 7). In control simulations, the *pf*-MLI synaptic strength was fine-tuned by LTP and LTD showing a prevalence of *pf*-MLI LTP in MLIs of which the activity went up and a prevalence of *pf*-MLI LTD in MLIs of which the activity decreased. In downbound PCs, the *pf*-PC synaptic strength rapidly decreased, with a net prevalence of LTD over LTP. In upbound PCs, the *pf*-PC synaptic strength rapidly increased due to LTP (LTD was absent due the lack of *cf* teaching signals in this microcomplex). In KO simulations, there were no apparent changes in *pf*-MLI LTD in MLIs that showed increases in activity, whereas *pf*-MLI LTP increased in these MLIs. In downbound PCs, there were no apparent changes in *pf*-PC LTD (enabled only in the MLI output KO), while LTP increased remarkably in *pf*-PC LTD KO and double KO. In upbound PCs, there were no apparent changes in *pf*-PC LTP in any of the KO conditions.

## 4. Discussion

This paper shows that plasticity in the MLI feedforward inhibitory circuit plays a crucial role in cerebellar learning during CEBC, complementing the role of *pf*-PC LTD to prevent potentiation of PC activity. These forms of plasticity impinging on PCs operate differentially in a network made of downbound and upbound microzones and microcomplexes, in which neuronal wiring and intrinsic discharge properties generate well-timed sequences of pauses and bursts. The convergence of the PC effects on $DCN_p$ neurons causes the behavioural response. These mechanisms were accounted for by a multiscale computational model, which was first validated against experimental data. KO simulations reproduced CEBC behaviour consistent with experiments from mice where genetic mutations conjunctively affected MLI feedforward inhibition and *pf*-PC LTD [25]. Then, the model was used to resolve the mechanistic link between microscopic neural circuit phenomena and sensorimotor learning in the cerebellum, explaining the impact of these genetic mutations on the underlying cerebellar neural mechanisms and predicting the behavioural outcome.

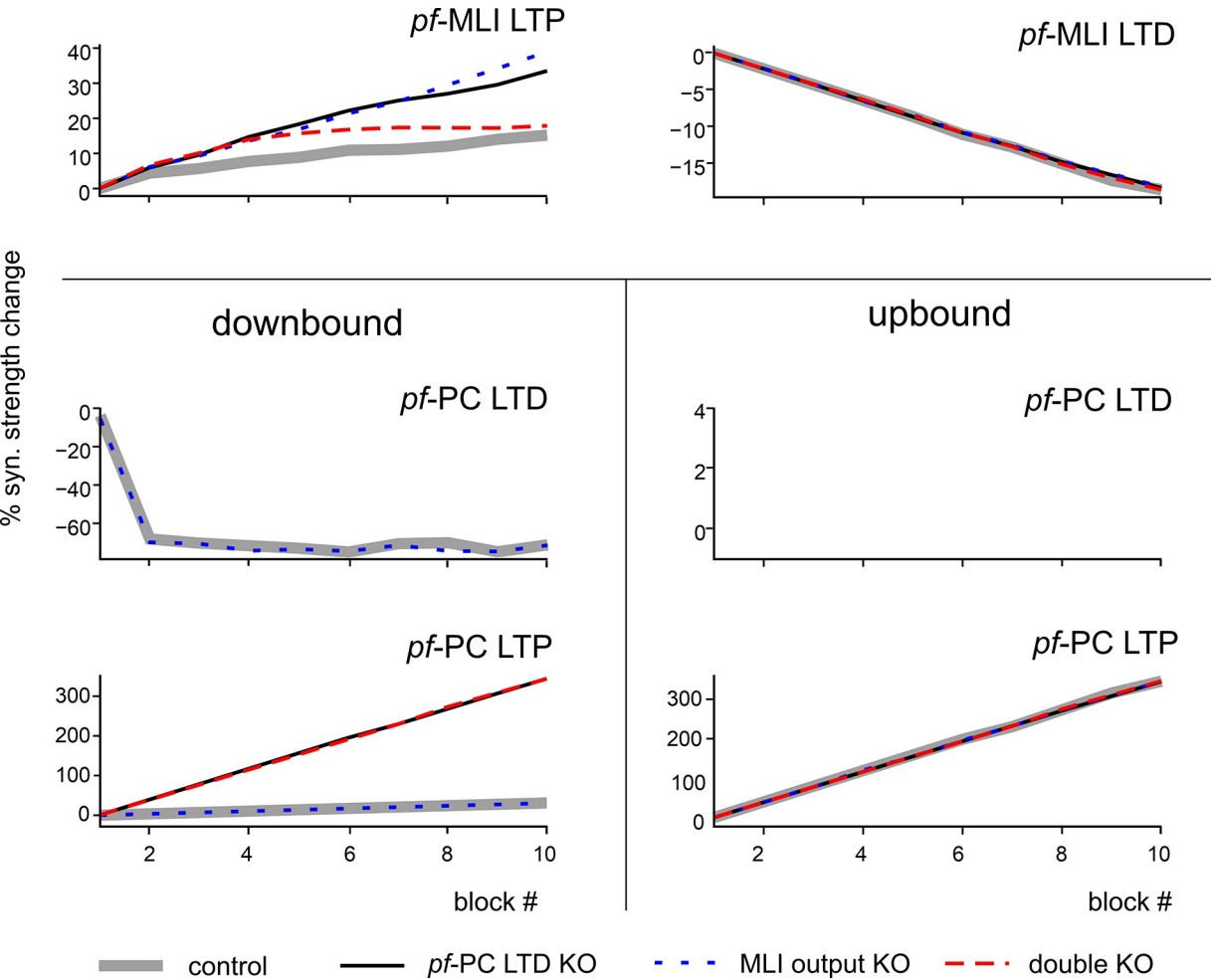

**Fig 7. Development of plasticity during learning in control and KO simulations.** The graphs show synaptic strength modulation (average % synaptic weight change with respect to initial weight, multiplied by the fraction of connections undergoing LTP or LTD) in the network plasticity sites in control and KO simulations. Plasticities at *pf*-MLI and *pf*-PC synapses of the downbound microcomplex are reported for each block, separating weights undergoing LTD and LTP.

## 4.1 The role of upbound and downbound microzones and microcomplexes

The importance of suppressing and facilitating PCs in distinct cerebellar microzones has recently been advocated for a basic task like the CEBC [26]. While the role of PC suppression in releasing DCN neurons has been shown in various labs [5,8,19,66], the role of facilitating PCs is still debated [26,28]. In synergy with suppressing PCs, facilitating PCs have been supposed to operate in push-pull controlling agonist and antagonist muscles (*orbicularis oculi* and *elevator palpebrae superioris*) to fine-tune the anticipated eyelid closure during CEBC [67]. This hypothesis has recently been confirmed by whole cell recordings of synaptic activity in DCN cells during eyeblink conditioning in vivo [57]. In the model comprising both the PCs and DCN, the net contribution of downbound and upbound microcomplexes is integrated to generate an optimal CR, supporting the possibility that they synergically control agonist and antagonist muscles driving eyelid closure. PCs in the model undergo different modulation of their firing rates: facilitation in the upbound microzone amplifies the response offset and thus the eyelid closure range, and suppression in the downbound microzone shapes the timing and

amplitude of the eyelid closure. From a different perspective, PCs in the model encode eyelid position, assuming it to be proportional to the net agonist/antagonist motor commands. Suppressing and facilitating PCs in downbound and upbound microzones could also be associated to eye closing and opening, respectively [19,27]. A similar linear readout of PC firing in downstream circuits has been proposed to explain the motor response in other timing tasks, like trace conditioning [68,69]. It should be noted that recent experiments in mice indicate that the activity of the downbound and upbound PCs correlate relatively well with eyelid velocity and position, respectively [28]. These results align with our data in that they agree with the impact of lesions on the downbound microcomplex, and that they are in line with the dynamics of movements with multiple degrees of freedom, where the downbound/upbound interplay may re-balance in case of lesions [26]. Interestingly, our simulations suggest that feedforward MLI inhibition operates gain regulation, while pf-PC LTD is particularly important for well-timed PC-SS suppression and spike-time precision. These observations resemble lesion and simulation results, in which suppressing and facilitating PCs endowed with LTD and LTP at pf-PC synapses play a key role in controlling accuracy and speed of saccadic eye movements [31,70]. In general, a double push-pull mechanism could contribute to increase response modulation (e.g., see Fig 3), balance learning and avoid saturation. In this context, LTP, which in effect may dominate in the upbound microzones, could regulate synaptic weight homeostasis emerging as a critical element of motor learning.

## 4.2 The role of plasticity in the inhibitory circuit

While pf-PC LTD was initially predicted by the Motor Learning Theory [11] and subsequently demonstrated experimentally [2,13,14], little was known about plasticity in the MLI inhibitory circuit [19,25,71]. Model simulations explain a series of experimental observations. First, the inhibitory branch passing through the MLIs can generate pauses in the high-frequency spontaneous firing of PCs [16,20,21], [72]. Second, the pauses are now predicted to play an important role not just for PC coding [73–75], but also for cerebellar circuit learning. Third, both plasticity of MLI feedforward inhibition and pf-PC LTD regulate the PC-SS output leveraging the double PC input from pfs and MLIs [23,24]. Finally, switching-off plasticity in the MLI feedforward inhibitory circuit affects CEBC just slightly (-15% reduction compared to control simulations). However, when plasticity in the MLI inhibitory circuit and pf-PC LTD are switched-off at the same time, CEBC is remarkably reduced (-50%), with residual learning being mostly explained by the CIO [25]. It should be noted that switching-off pf-PC LTD slightly reduces the learning rate, but that, at the end of training, the gain converges toward control values (-10%). Therefore, the two plastic mechanisms are not equivalent and contribute non-linearly, yet synergistically, to CEBC (i.e., their combined effect is larger than their sum). Apparently, the effect of KO of MLI feedforward inhibition seems larger in biological recordings than in simulations, suggesting that some mechanistic aspects may not be accounted for by the model.

## 4.3 Model assumptions

The large number of trials required to simulate plastic changes during CEBC learning in mice and the jump to mesoscale (cerebellar cortex associated with DCN and IO embedded in a controller) required some simplifying assumptions. Nonetheless, these were purposefully designed to keep the biological plausibility of simulations, considering the task and the mechanisms within the micromodules under investigation.

The neurons were single-point but maintained salient discharge properties, e.g., linear I-f relationship, burst/pause and rebound responses [55]. Since the CEBC input is "mono-dimensional" (a tone/LED light) and not time-varying, sparse and non-recurrent encoding in the

granular layer was used to discriminate the time instants of the CS at the Purkinje layer level [10,11,48,59,76], in absence of dendritic integration.

Recent studies elucidated the role of PC dendritic processing in cerebellar computation and learning: differentiated excitability of PC dendritic tree makes active inputs heterogeneous and contribute to pattern separation, and dendritic excitability can be plastic [77–79]. This is crucial also for PC responses to *cf* inputs that are the instructive signal for CEBC: clustered activation of the dendritic tree from *pf* and MLI inputs causes distinct responses to *cfs* [80]. This can involve multiple *cf* innervations of the PC dendritic tree especially in the cerebellum of human and higher non-human primates [81]. Differentiated excitability and plasticity of PC dendrites could be an additional mechanism contributing to CEBC, together with synaptic plasticity, as shown in [82] where a lack of dendritic excitability leads to reduced CEBC. This could be investigated in future implementations of the current SNN model, where PCs are represented as multi-compartmental neurons with dendritic processing instead of point neurons. Adding dendritic computation will be crucial to simulate more complex tasks, where the cerebellum receives multi-modal and time-varying inputs.

Simulations reproduced long-term plasticity mechanisms: absence of short-term plasticity is a minor issue, since CEBC is "single jerk" and does not involve dynamic evolution of signals along a motor sequence.

The assumption that conditioned response timing is encoded in the granular layer derives from spiking models previously used for CEBC [33,48]. Different granule cells spike patterns were associated with different instants of CS, representing the passage of time. Thanks to the high number of GrCs and the inhibitory GoC loop, a sparse non-recurrent pattern of activity was ensured during presentation of the CS [59]. The model included electrical synapses and gap-junctions between GoCs [83], which are supposed to control the timing of feedforward inhibition in the granular layer [84]. Other mechanisms, like granular layer plasticity [85,86], as well as recurrent collaterals from DCN neurons into the granular layer [87], may intervene in more complex tasks, where multiple sensory signals with different time scales are combined [14,88].

Experimental studies show that the CIO emerges in the ISI during learning [19,89,90] and suggest that it originates from CS-related signals in extra-cerebellar nuclei (e.g., the IO) [47]. Here, the CIO was modelled as an input to the IO and was modulated by feedback inhibition from $DCN_{GABA}$.

Our study focused on synaptic plasticity for CEBC in the cerebellar cortex, but additional plasticity mechanisms could contribute to the conditioned response learning. Plasticity of neuron intrinsic excitability has been shown recently to play a role in CEBC. Specifically, PC pauses can be reinforced by intrinsic electroresponsive mechanisms [72] and intrinsic plasticity, which can reduce spontaneous PC firing even without increased inhibitory input [8,91,92]. In addition, intrinsic excitability of dendrites in MLIs could contribute to increased MLI feedforward inhibition [93]. Preliminary simulations with altered intrinsic excitability show decreased or slowed-down CEBC (S8 Fig), due to an unbalance of facilitation and suppression mainly in downbound PCs: when PC baseline activity is lower, they are more sensitive to synaptic potentiation, resulting in facilitation of their activity at the end of learning [26], which prevents proper CEBC; when baseline activity is higher, depression still occurs but it takes longer to release the DCN, leading to slowed-down CEBC. Despite these promising preliminary results, the role of intrinsic plasticity mechanisms contributing to conditioned response timing remains to be evaluated [94–96] and should be considered in future implementations of the model. Experimental evidence supports a role for plasticity and recurrent connection loops also in the cerebellar nuclei. Plasticity at $PC-DCN_P$ and *mf*-$DCN_P$ connections has been measured in rodents and rabbits [97–100]. In addition, feedback loops from DCN to the granular

layer can amplify the effect of learning [87]. The absence of nuclear plasticity is a minor issue here, where learning occurs over the fast time scale of cortical plasticity. Nuclear plasticity should become critical in more complex associative learning protocols involving memory consolidation and saving [35,36,101,102].

Generalizing the current model to simulate more complex behaviours would require extending spatiotemporal representations to multiple scales, e.g., by introducing dendritic processing, synaptic dynamics, additional plasticity rules, and DCN and PC wiring loops [14,45,87,103–107], as well as consolidation of the synaptic plasticity by sealing of the perineuronal nets [108].

## 4.4 Conclusions

Simulations predict the neural underpinning of CEBC and address the core of cerebellar computation [3]: (i) the downbound and upbound microzones cooperate in generating the output, but the downbound microzone (thanks to its unique *cf* input) is most critical for error-based learning, (ii) plasticity in the MLI feedforward inhibitory circuit cooperates with *pf*-PC LTD, (3) redundancy of mechanisms (*pf*-PC LTD, plasticity of MLI feedforward inhibition, CIO-related neuronal responses) allows partial compensation of lesions, (4) feedforward MLI inhibition operates gain regulation, while *pf*-PC LTD is particularly important for time-precision of PC-SS suppression, (5) burst-pause patterns in PCs and pause-burst patterns in DCN neurons within the same microcomplex can be regulated by synaptic plasticity. These mechanisms together with those occurring in the IO establishing the micromodules make the circuit able to predict the precise timing of correlated events, to finetune adaptive associative behaviours, and to compensate the effect of lesions. Although CEBC is an elementary associative response, it can be considered a kernel of cerebellar functioning and the results obtained here may also apply to more complex behaviours, including object manipulation and cognitive and emotional processing.

## Supporting information

**S1 Fig. Model tuning, validation, and prediction.** Block diagram of the relationship between experiments and simulations. Electrophysiological data have been used to tune the microcircuit mechanisms and make then compatible to the in-vivo recordings of reference. The tuned model was able to generate a physiological CEBC learning curve. Then the model was re-tested after changing some mechanisms mimicking the alterations induced by specific genetic mutations (KO). The model straightforwardly simulated the pathological learning curves providing a solid element of validation. The model was then used to predict the underling changes in neuronal network dynamics.
(TIF)

**S2 Fig. CEBC protocol and neural activity analysis.** The CEBC protocol is organized in 10 blocks made of 100 trials each. A trial consists of 500 ms before and 500 ms after the CS-US pairing period, which lasts 260 ms (0–260 ms CS, 250–260 ms US). In a trial, spike are recorded from each neuron and the spike pattern is stored. Then, each timeseries is processed by extracting the activity curve as spike density function (SDF). The SDF modulation is computed as the activity ratio in the CR window with respect to the corresponding baseline. The example shows the SDF for PCs, MLIs and DCN$_p$ neurons.
(TIF)

**S3 Fig. Model parameter tuning against experimental data.** (A) Kernel of the *pf*-MLI plasticity rule. The experimental data (orange line) show the modification of MLI activity during

learning [19]. The model (black line) has an alpha-shaped kernel for *pf*-MLI LTP triggered by *cf* spikes (purple bar). (B) The bar plots (mean±s.d.) shows the baseline firing rate of MLI, downbound PC and downbound DCN$_p$ in control experimental recordings compared to simulations.
(TIF)

**S4 Fig. Simulations with different ISIs. (A)** Motor output and %CR in the 200 ms (top) and 300 ms (bottom) ISI simulations. Both motor output evolution and %CR increase throughout learning blocks, as in the simulations with ISI = 250 ms. **(B)** Onset and peak timing of the conditioned response. The motor output timing is adjusted to the US, which occurs at different ISIs in the different simulation conditions.
(TIF)

**S5 Fig. Spiking activity of representative single neurons.** Raster plots of spiking activity of exemplar MLI, PC and DCN$_p$ neurons during CEBC learning in control simulations. The spikes cover 10 blocks of 100 simulations each. Firing rate changes across blocks are summarized in the right graphs (% firing rate change with respect to the first block)
(TIF)

**S6 Fig. Activity of neuronal populations: downbound and upbound microcomplexes in control simulations.** SDF traces undergoing significant suppression or facilitation are shown normalized to baseline for the downbound microcomplex (PC and DCN$_p$ and downbound-preferring MLIs) and upbound microcomplex (PC and DCN$_p$ and upbound-preferring MLIs) in the last (10$^{th}$) block of in control simulations. In all panels, SDF traces are normalized to baseline, and vertical lines indicate CS and US (cf. Fig 1). In each population, SDF traces are averaged (thick lines). The boxplots of % SDF modulation compare simulation and experimental value distributions.
(TIF)

**S7 Fig. Activity of neuronal populations: downbound microcomplex in control and KO simulations.** SDF traces undergoing significant suppression or facilitation are shown normalized to baseline for the downbound microcomplex (PC and DCN$_p$ and downbound-preferring MLIs) in the last (10$^{th}$) block in control, *pf*-PC LTD KO, MLI output KO, double KO simulations. In all panels, SDF traces are normalized to baseline, and vertical lines indicate CS and US (cf. Fig 1). In each population, SDF traces are averaged (thick lines).
(TIF)

**S8 Fig. Simulations with altered intrinsic excitability of PCs.** (A) Motor output with decreasing (top) and increasing (bottom) PC spontaneous firing: both the motor response and the learning curve are reduced or slowed down. (B) A different pattern of facilitation and suppression at the end of learning is observed in downbound PCs, where the unbalance between the two mechanisms causes an insufficient release of the DCN and thus the observed alteration of learning.
(TIF)

**S1 Text. Table A.** *Neurons in the network.* Number of elements in the olivocerebellar network (downbound and upbound neuron numbers in brackets), and model parameters for each neuron type, from [55]. **Table B.** *Connectivity in the network.* Connection types in the olivocerebellar network are reported together with the global average convergence and divergence ratios, and synaptic parameters, i.e. weight and delay. Excitatory and inhibitory connections are reported with positive and negative synaptic weight, respectively. When differentiated, downbound and upbound weights are reported (downbound; upbound). **Equation A.** *pf-PC*

*plasticity model.* Learning rule for the pf-PC plasticity site, derived from previous works. (DOCX)

## Acknowledgments

We thank the PizDaint support team at the CSCS Swiss National Supercomputing Centre for continuous informatic advice and support.

For HPC resources, we acknowledge the use of Fenix Infrastructure resources, which are partially funded from the European Union's Horizon 2020 research and innovation programme through the ICEI project under the grant agreement No. 800858, through the specific project ich027 - "Simulations of cerebellar circuits: from detailed models to large-scale spiking networks with plasticity".

## Author Contributions

**Conceptualization:** Alice Geminiani, Claudia Casellato, Henk-Jan Boele, Alessandra Pedrocchi, Chris I. De Zeeuw, Egidio D'Angelo.

**Data curation:** Alice Geminiani.

**Formal analysis:** Alice Geminiani.

**Funding acquisition:** Henk-Jan Boele, Alessandra Pedrocchi, Chris I. De Zeeuw, Egidio D'Angelo.

**Investigation:** Henk-Jan Boele.

**Methodology:** Alice Geminiani, Claudia Casellato.

**Software:** Alice Geminiani.

**Supervision:** Chris I. De Zeeuw, Egidio D'Angelo.

**Visualization:** Alice Geminiani, Claudia Casellato.

**Writing – original draft:** Alice Geminiani, Claudia Casellato, Egidio D'Angelo.

**Writing – review & editing:** Henk-Jan Boele, Alessandra Pedrocchi, Chris I. De Zeeuw.

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
