## [Decision Letter · Decision Letter 0]

23 Oct 2023

Dear Dr. Geminiani,

Thank you very much for submitting your manuscript "Mesoscale simulations predict the role of synergistic cerebellar plasticity during classical eyeblink conditioning" for consideration at PLOS Computational Biology.

We are sorry for the delay in making this decision, which was due to a large number of invited reviewers either declining or failing to respond. We have only obtained one review to date and so to avoid further delay, I have carried out a detailed review myself, with the comments appended below.

In light of the reviews (below this email), we would like to invite the resubmission of a significantly-revised version that takes into account the reviewers' comments. Note that these highlight that although the work appears of substantial interest, the model description is insufficiently detailed, and the explanation of several key mechanisms is unclear. They also note that the provided link to code did not appear to work.

We cannot make any decision about publication until we have seen the revised manuscript and your response to the comments. Your revised manuscript is also likely to be sent to reviewer for further evaluation.

Sincerely,

Barbara Webb

Academic Editor

PLOS Computational Biology

Marieke van Vugt

Section Editor

PLOS Computational Biology

Reviewer's Responses to Questions

**Comments to the Authors:**

Reviewer #1: CEBC has been well-studied by many researchers, including Drs. Alkon, Mauk, Thompson, de Zeeuw, and Hesslow. While there are agreement and disagreement among research groups (e.g., if synaptic plasticity is involved in the learning), this manuscript answers an important question of how the synergy of MLI inhibitory synaptic transmission and excitatory synaptic plasticity is involved in the CEBC learning process. The author built the in silico cerebellar system using a supercomputer with a high temporal resolution and challenged the motor learning theory.

The conclusion was plausible in their relatively simplified system, and the resultant message closely aligned with previous experimental outcome from coauthors of Prof. de Zeeuw’s group (BoeleH De ZeeuwCI 2018sciadv).

My questions and comments are following:

1.

I wonder the intrinsic excitability of neurons is a critical factor, which may change the simulation outcome. I would like to ask the result as an independent figure when PC excitability is set high (e.g., basal firing frequency rate to 150% increase) and low (e.g., basal firing rate to 60% decrease) in both upbound and downbound PCs.

2.

Schreurs and Alken (1991; 1997)(Schreurs et al., Brain Res. 1991 doi: 10.1016/0006-8993(91)91100-f. PMID: 1868333; Schreurs et al., J Neurophysiol. 1997 doi: 10.1152/jn.1997.77.1.86. PMID: 9120599) suggested an increase in the PC firing frequency during CEBC and excitability of dendrites. However, the manuscript merely discuss:

“655 The neurons were single-point but maintained salient discharge properties, e.g., non-linear I-f

656 relationship, burst/pause and rebound responses [56]. The absence of dendritic integration is a minor

657 issue, since the CEBC input is “mono-dimensional” (a tone) and does not require dendritic integration

658 over an extended multiparametric space. …

I consider PC dendrites more complex than the authors expect, and they have both dendritic integration, branch-specific compartmentalization, and branch-specific enhancement (Zang et al./De Schutter 2018 Cell Rep; Ohtsuki et al./ Hansel, 2012 Neuron; Busch & Hansel, 2023 Science). Therefore, the claim of this manuscript may miss a biological significance. One of the researchers found that the difference of the branch excitability confers the branch-specific information processing of synaptic transmission, too (Ohtsuki, 2020 JNS). I wish the authors to describe the comment on it with referring citations as a paragraph. In fact, L7-SK2 mice show impaired CEBC (Grasselli, Boele, / Hansel 2020 PLoS Biol). Thus, this manuscript will be established if it append the discussion as for another hypothesis or prediction. L.655-658 should be rephrased.

Zang Y, Dieudonné S, De Schutter E. Cell Rep. 2018;24(6):1536-1549. doi: 10.1016/j.celrep.2018.07.011. PMID: 30089264

Ohtsuki G, Piochon C, Adelman JP, Hansel C. Neuron. 2012;75(1):108-20. doi: 10.1016/j.neuron.2012.05.025. PMID: 22794265

Busch SE, Hansel C. Science. 2023;381(6656):420-427. doi: 10.1126/science.adi1024. PMID: 37499000

Ohtsuki G. J Neurosci. 2020;40(2):267-282. doi: 10.1523/JNEUROSCI.3211-18.2019. PMID: 31754008

Grasselli G, Boele HJ, Titley HK, Bradford N, van Beers L, Jay L, et al. PLoS Biol. 2020;18(1):e3000596. doi: 10.1371/journal.pbio.3000596. PMID: 31905212

3.

Related to the question above, Prof. Sang Jeong Kim’s group recently found the LTD of intrinsic excitability in animals (Shim et al. /Kim 2017JNS doi:10.1523/JNEUROSCI.3464-16.2017 pmid:28495974), which potentially reduces PC firing frequency without inhibitory conductance. I wish the authors to ask the possibility of the plasticity in their model as discussion.

4.

In the model, I wonder about the conductance of MLI GABAergic projection. How many projections of stellate cells and basket cells are assumed? And how much conductance is given in the authors’ model? I ask the authors to provide every parameter of synaptic conductance and basic property of each neuron, as an additional table.

Given LTP of pf-MLI synapse or LTP of MLI intrinsic excitability was induced, is the +321%+/-325 facilitation (Table 1) plausible? I’ve never heard of it in vivo and in vitro, to my knowledge. Some more complementary mechanisms may be inherited (cf. a synergy of intrinsic excitability of dendrites and synaptic transmission Zhang and Linden, 2003 https://doi.org/10.1038/nrn1248).

5.

Data and Code Availability

https://github.com/AliceGem/cereb_scaffold_ebc

does not work.

6.

l. 124-126 was not understandable for me.

124 Furthermore, complex spikes were extracted from PC in vivo recordings: throughout learning, a

125 complex spike response emerged during the ISI (conditioned complex spike), with 43% probability

126 of occurrence in fully trained mice, at an average latency of 88 ms after CS onset.

7. Abbreviation.

l. 246, Spike Density Function

l. 312, spike density function (SDF)

8.

Correct? Which one?

l. 352-353

see also Fig. 4 below

9.

Figure 4C

What do asterisks mean by?

10.

l.573-582

Description on Fig.7 is missing.

********

Editorial review 

This paper extends previous work on large-scale spiking models of
cerebellar circuits
in the context of classical eyeblink conditioning (CEBC). The key new features are the inclusion of
two cerebellar modules with different properties and plasticity in the feedforward inhibitory loop, in addition to the parallel fibre to Purkinje cell plasticity that has been the focus of most previous
models. The simulation demonstrates that these plasticity mechanisms interact such that only their combined absence has a significant effect on learning, consistent with experimental results from mice using mutations that target one or other or both systems. 

Overall this is an interesting study that provides insight from an anatomically and physiologically constrained model into cerebellar function. This includes reproducing the genetic knockout effects and providing some predictions of the expected changes in microcircuits during learning, i.e., which cells should show significant activity changes during learning. There is an interesting discussion of the insights provided by the model as well as additional factors that might be relevant to include in future modelling. 

However
a number of revisions are needed to improve the clarity of the paper.  

Line 26 “prompts to remap” could be better expressed. 

27-28 first mention of ‘upbound’ and ‘downbound’ modules. For a reader not highly familiar with the cerebellar system it remains
rather unclear throughout most of the paper what these terms refer to, so some explanation is needed earlier. 

38-39 “fine tuning of adaptive associative behaviours at a high spatio-temporal resolution” - this indeed seems like it should be the most interesting outcome of the presented work, in particular that this might explain why there are multiple forms of plasticity in a circuit that ultimately is
learning a rather simple task. However, it is not sufficiently demonstrated in this paper that the circuit performs ‘fine tuning’,
e.g., there is no exploration of how the circuit performs when details of training, such as ISI, are varied.  

45 It seems odd to emphasise here the co-termination of the CS and US as the defining feature of the CEBC paradigm, whereas the critical factor for generating a “well-timed” CR is the temporal relationship between the onset of the CS and US, not their termination. 

63-66 I found this unclear, why would PC firing be reduced by an increase in the pf input? Isn’t the hypothesis that pf-PC LTD decreases the pf input, and hence the PC firing? And in what sense can PC firing “hardly be reduced” by a change in pf input? 

70-72 why does the role of pf-PC LTD “to prevent potentiation” not suffice to bring SS activity of PC down? 

78-80 Again to a reader unfamiliar with the cerebellum, it is entirely opaque what is being referred to for “zebrin-negative” and “zebrin-positive” 

140-144 This seems a very brief explanation of the modelling approach, is it sufficiently covered in the cited references? 

147-162 this seems like material that belongs in the introduction. 

170-193 This description of the model has insufficient detail. E.g. it provides the total number of cells but not the number of each cell type, for ii) and iii) not even the number of cells. Just listing the connection types seems inadequate. 

203 – what are ‘parrot’ neurons? 

207 describes the parameters as being ‘set’ to reproduce physiological data. How was this done? Extracted directly from data?
Hand tuning? Parameter search? Some automated fitting method? 

Lines 216-239 The learning rule needs clearer explanation. It is described as STDP, but STDP is normally understood to refer to change in a synapse due to the relative timing of spikes in the presynaptic and postsynaptic neurons, with respect to that synapse. However here the change in pf-PC or pf-MLI (so pf is the presynaptic neuron and PC/MLI the postsynaptic neuron) depends on the relative timing of the spike in cf and pf. The text does not mention any dependence on MLI activity, yet in the equations this is a condition on the occurrence of LTP or LTD. It might also help in the equations to use ‘cf’ rather than ‘IO’ (or to refer to IO activity rather than cf activity in the preceding text) to make the connection between these terms explicit. 

After reading further it is mentioned on line 385-386 (fig 1 caption) that the IO activity in cfs elicits a complex spike in PC and activates MLI; so I am inferring that the assumed mechanism here is that cf spikes cause (complex spike?) activity in PC or MLI, and this is the ‘postsynaptic’ activity that makes it an STDP learning rule?
But the implemented equations use the cf spike, not the MLI spike, for the timing. 

241 Again please explain how the parameters were ‘set’. 

262-266 This is not at all well described, but the impression provided is that the CIO was manually manipulated to change across training trials, in a way that mimics learning, but without this actually being an emergent property of the circuit or the use of any actual learning rule used to induce the change. It was not clear how the ‘complex spike’ fits into the overall system but it seems to be key later on (e.g. line 559) to explain residual learning after other factors are removed. On lines 425-428 it is described as a ‘result’ of the model that the “complex spike modulation during learning showed a typical evolution” but it seems this feature was directly built into the model. Please explain this better. 

409- 412 I missed anything in the
earlier model description explaining
this differential innervation of microzones by the MLI. 

601 “predicting the impact of genetic mutations” - given these were known, and a focus in model construction, it seems too strong to say the model predicted these effects, rather, it was consistent with these observations. 

**Have the authors made all data and (if applicable) computational code underlying the findings in their manuscript fully available?**

Reviewer #1: **No: **Data and Code Availability

https://github.com/AliceGem/cereb_scaffold_ebc

does not work. Maybe, error?

PLOS authors have the option to publish the peer review history of their article (what does this mean?). If published, this will include your full peer review and any attached files.

Reviewer #1: No
---

## [Decision Letter · Decision Letter 1]

12 Feb 2024

Dear Dr. Geminiani,

We are pleased to inform you that your manuscript 'Mesoscale simulations predict the role of synergistic cerebellar plasticity during classical eyeblink conditioning' has been provisionally accepted for publication in PLOS Computational Biology.

Best regards,

Barbara Webb

Academic Editor

PLOS Computational Biology

Marieke van Vugt

Section Editor

PLOS Computational Biology

Reviewer's Responses to Questions

**Comments to the Authors:**

Reviewer #1: The author have addressed all my concerns. I have no more comments.

This is an outstanding simulation study on cerebellar EBC learning. Well done!

**Have the authors made all data and (if applicable) computational code underlying the findings in their manuscript fully available?**

Reviewer #1: Yes

PLOS authors have the option to publish the peer review history of their article (what does this mean?). If published, this will include your full peer review and any attached files.

Reviewer #1: **Yes: **Gen Ohtsuki

---

## [Editor Report · Acceptance letter]

27 Mar 2024

PCOMPBIOL-D-23-00960R1 

Mesoscale simulations predict the role of synergistic cerebellar plasticity during classical eyeblink conditioning

Dear Dr Geminiani,

I am pleased to inform you that your manuscript has been formally accepted for publication in PLOS Computational Biology. Your manuscript is now with our production department and you will be notified of the publication date in due course.

With kind regards,

Anita Estes
